# Cluster Analysis of Health-Related Lifestyles in University Students

**DOI:** 10.3390/ijerph17051776

**Published:** 2020-03-09

**Authors:** Miquel Bennasar-Veny, Aina M. Yañez, Jordi Pericas, Lluis Ballester, Juan Carlos Fernandez-Dominguez, Pedro Tauler, Antoni Aguilo

**Affiliations:** 1Department of Nursing and Physiotherapy, Balearic Islands University, Cra. de Valldemossa, Km 7.5, 07122 Palma, Illes Balears, Spain; miquel.bennasar@uib.es (M.B.-V.); jordi.pericas@uib.es (J.P.); jcarlos.fernandez@uib.es (J.C.F.-D.); aaguilo@uib.es (A.A.); 2Research Group on Global Health & Human Development, Balearic Islands University, Cra. de Valldemossa, Km 7.5, 07122 Palma, Illes Balears, Spain; 3Department of Specific Didactics and Pedagogy, Educational and Social Research and Training Research Group, Balearic Islands University, Cra. de Valldemossa, Km 7.5, 07122 Palma, Illes Balears, Spain; lluis.ballester@uib.es; 4Research Group on Evidence, lifestyles and Health Research, Instituto de Investigación Sanitaria Illes Balears, Cra. de Valldemossa, Km 7.5, 07122 Palma, Illes Balears, Spain; pedro.tauler@uib.es

**Keywords:** lifestyle, clustering, risk factors, university students, physical activity, diet, smoking, alcohol, mediterranean diet, health-related behaviors

## Abstract

Health-related lifestyles in young adults are a public health concern because they affect the risk for developing noncommunicable diseases. Although unhealthy lifestyles tend to cluster together, most studies have analyzed their effects as independent factors. This study assessed the prevalence, association, and clustering of health-related lifestyles (smoking, alcohol consumption, physical activity, and quality of diet) among university students. This cross-sectional study examined a sample of student participants from the University of the Balearic Islands (n = 444; 67.8% females; mean age: 23.1 years). A self-reported questionnaire was used to assess health-related lifestyles. Men that consumed more alcohol, had less healthy diets, were more likely to be overweight, and performed more physical activity. Women had a higher prevalence of low weight and performed less physical activity. Physical activity had a negative association with time using a computer (OR: 0.85; 95% CI: 0.76, 0.95) and a positive association with adherence to the Mediterranean diet (OR: 1.16; 95% CI: 1.02, 1.32). Adherence to the Mediterranean diet had a negative association with tobacco consumption (OR: 0.52; 95% CI: 0.30, 0.91), and positive associations with having breakfast every day (OR: 1.70; 95% CI: 1.05, 2.76) and consuming more daily meals (OR: 1.43; 95% CI: 1.10, 1.87). Cluster analysis indicated the presence of three distinct groups: Unhealthy lifestyles with moderate risk; unhealthy lifestyles with high risk; and healthy lifestyles with low risk. Health promotion interventions in the university environment that focus on multiple lifestyles could have a greater effect than interventions that target any single lifestyle.

## 1. Introduction

Previous research reported that modifiable unhealthy lifestyles, such as smoking, poor diet, physical inactivity, and alcohol consumption, independently contributed to increased premature morbimortality and development of noncommunicable diseases [1,2,3]. The prevalence of unhealthy lifestyles is especially high in young adults, including university students. The global epidemic of overweight and obesity is an important public health concern because it may lead to increases in cardiovascular diseases, metabolic conditions such as type 2 diabetes, and cancer [4]. 

Lifestyles that begin during adolescence tend to consolidate during youth (when attending a university) and continue throughout adulthood [5]. When an individual enters a university, important changes occur, such as increased individual autonomy, exposure to a new environment and social networks, and less parental control, that can lead to psychological stress [6,7]. Psychological stress is associated with alcohol and tobacco consumption, physical inactivity, and low consumption of fruits and vegetables [5,8,9]. These unhealthy lifestyles, together with socio-economic factors, could negatively impact university students [10]. Other research has suggested an association of health status with sociodemographic, economic, and psychosocial factors [11]. Furthermore, an individual’s perceived state of health is a multifactorial indicator that provides information about the individual’s physical and mental health, and is considered a good predictor of morbidity, mortality, quality of life, and well-being [12,13,14].

Previous studies have evaluated the effects of several single health-related risk factors, but no studies have considered these factors as clustered lifestyles, nor their synergistic effects, in the university environment. However, a few studies reported that unhealthy lifestyles tended to associate with each other [15,16,17,18]. The presence of such clustering could have significant implications for studies of morbidity and mortality, and for the design of preventive and health promotion interventions, because it means that interventions should focus on the whole lifestyle rather, than a single factor [19,20].

To our knowledge, only a few studies have analyzed the relationships of psychological stress, self-perception of health status, quality of life, and multiple health-related lifestyles in university students. The aim of the present study was to assess the prevalence, association, and clustering of major health-related lifestyles—smoking, alcohol consumption, physical activity, and diet quality— with stress, health status, and quality of life among university students. 

## 2. Materials and Methods 

### 2.1. Study Design

This cross-sectional study initially examined a sample of 473 Spanish students from the University of the Balearic Islands. All subjects were 18 to 40 years-old and were enrolled at the university for more than one year. Twenty-nine participants were excluded due to submission of incomplete surveys or questionnaires that were incorrectly completed. Therefore, data from 444 students were used for analysis.

### 2.2. Sample and Procedure

Participants were recruited using a randomized, multistage, conglomerate procedure, and stratified by an academic major (art and humanities, sciences and health sciences, social and law sciences, and engineering), with the proportion of overall participants recruited in a given academic major being equal to the proportion of students in that major at the university. The randomization unit was academic major in the first stage and academic year in the second stage. The gender, age, and academic major of study participants were similar to those of students overall at the university.

### 2.3. Measures

The survey was developed from a short version of the Spanish National Health Survey [21] in combination with other validated questionnaires [14,22,23,24], was self-administered with the supervision of trained interviewers, and examined the following parameters.

Demographic and anthropometric characteristics. Age, gender, academic major and year, and self-reported weight and height were collected. Body mass index (BMI) was calculated as kg/m^2^ and BMI groups were used to classify participants following World Health Organization criteria: Underweight (BMI < 18.5 kg/m^2^), normal weight (BMI = 18.5 to 24.9 kg/m^2^), overweight (BMI = 25.0 to 29.9 kg/m^2^), and obese (BMI ≥ 30.0 kg/m^2^) [25].

Health status, quality of life, and stress. Self-assessment of health status was used as a global health indicator [14], and self-perceived health status was obtained by the answer to the question: “Within the last 12 months, how do you rate your overall health status?” The five possible answers were “very good”, “good”, “fair”, “bad”, or “very bad”. The participants were categorized into two groups for descriptive analysis; students in one group answered “fair”, “bad”, or “very bad” and students in the other group answered “good” or “very good”. Several studies have shown good test-retest reliability of self-reported health measures, even better than in assessments of more specific health issues [26,27]. Questions about self-perceived quality of life and level of stress, using a Likert scale from 1 (very low) to 5 (very high), were also administered.

Social class. Social class was ascertained from highest parental occupation using the Spanish adaptation of the Goldthorpe classification suggested by the Spanish Society of Epidemiology [28]: Class I (upper class) includes executives, managers, and university professionals; Class II (middle class) includes intermediate occupations and employees; and Class III (lower class) includes manual workers. A two-category classification (blue-collar vs. white-collar) was also used for comparisons.

Smoking. Participants were asked if they smoked, and were then classified as smokers, nonsmokers, or former smokers. Smokers were asked about the number of cigarettes consumed per day, and whether they were occasional or frequent/regular smokers. 

Drinking. The consumption of alcohol (yes vs. no), frequency of alcohol consumption, and frequency of drunkenness, were recorded. The types and amounts of alcoholic drinks were recorded to calculate the total amount of ethanol (g) consumed.

Drug consumption. Drug consumption was quantified by the answer to the following question: “How often do you consume any of the following substances: Tranquilizers, cannabis, cocaine, ecstasy, hallucinogens, amphetamines, or heroin?” The possible answers were “never”, “occasionally”, “weekly”, or “daily” for each of three time periods (“anytime in your life”, “within the last 12 months”, and “within the last 30 days”).

Physical activity and sedentary habits. The standard form of the International Physical Activity Questionnaire (IPAQ) was used to determine physical activity (PA) levels and total weekly time of PA [22]. IPAQ results are expressed as metabolic equivalents (METs)-minutes per week. The physically active participants were also asked about the type of PA they performed, and the weekly time spent performing a PA. Questions about the daily time spent in sedentary activities (watching TV, using a computer, and doing homework) were also in the survey.

Dietary habits and diet quality. Questions were asked to determine the number of meals eaten per day and the frequency of eating breakfast (“every day”, “sometimes”, “almost never”, or “never”). Adherence to the Mediterranean diet (MD) was evaluated using the PREDIMED index and the Mediterranean Diet Score (MDS). The PREDIMED index assesses the frequency of consuming 14 items [23] that are indicative of high adherence (olive oil, fruits, legumes, nuts, wine, seafood, and poultry) and the consumption of foods indicative of low adherence (red or processed meats, commercial bakery items, sugary drinks, and sweets). Each item was scored as 0 or 1, and a global score of 9 or higher indicated acceptable adherence [23]. A modified version of the MDS that evaluated the consumption of eight typical (cereals, whole grain cereals, legumes, fruits, nuts, vegetables, fish products, and olive oil) and two atypical (dairy products and meat products) MD food groups was also used. This score is a variation of the original MDS, which was adapted to consider the dietary patterns and nutritional needs of this population group [24,29]. The MDS was categorized as indicating high adherence (MDS ≥ 5) or low adherence (MDS < 5).

### 2.4. Statistical Analysis

The listwise deletion method (with complete case deletion) was used to account for missing data [30]. Descriptive analysis was used to report the frequency and distribution of categorical variables, and means and standard deviations (SDs) were reported for quantitative variables. 

Bivariate association analysis was performed using the χ^2^ test, with correction using Fisher’s exact statistic when required. Student’s t-test for independent samples was used to compare means.

To evaluate the association of different risk factors with MD adherence (high vs. low) and weekly PA (yes vs. no), two adjusted logistic regression models were used. All variables that were considered relevant (age, sex, social class, BMI, etc.) and those with a *p*-value below 0.25 in the univariate analysis were included in the adjusted model. Each variable was extracted (one-by-one) according to the level of significance in the model. Possible interactions and their statistical significance were evaluated, and possible collinearity was determined. The goodness of fit of the model was assessed using the Hosmer-Lemeshow C statistic and calculation of the area under the receiver operating characteristic (ROC) curve [31].

The cluster analysis used the k-means algorithm to identify groups of students who shared common health-related lifestyles based on seven factors. This method identifies uniform groups by maximizing inter-group variance and minimizing intra-group variance. This method is less influenced by the presence of atypical cases than other clustering methods [32]. The scaling of variables is a fundamental criterion, so all variables were standardized before the cluster analysis. Once the variables were selected, the grouping process was performed to identify clusters using the criteria of “largest number of possible groups and largest number of cases explained”.

Statistical analyses were performed using the IBM SPSS Statistics version 24 (SPSS/IBM, Chicago, IL, USA) and STATA version 11 (StataCorp LP, TX, USA). In all cases, a *p*-value below 0.05 was considered significant.

### 2.5. Ethical Aspects

The study protocol was approved by the Institutional Review Board of the Balearic Islands Health Service (CEI-IB Ref. No.: 1685/11). All participants gave written informed consent.

## 3. Results

### 3.1. General Characteristics of Participants

The 444 student participants included 301 females (67.8%) and 143 males (32.2%). The mean age was 23.1 ± 5.7 years, and most students had normal BMI values and were in the middle class (Table 1).

### 3.2. Health-Related Lifestyles of Participants

Table 2 shows the health-related lifestyles of the participants. About 89% of them reported a good quality of life and 91.0% of them reported a good health status. However, 50.6% of them reported high or very high levels of stress, and women were more stressed than men (86.9% vs. 69.9%; *p* < 0.001). Age and social class were unrelated to stress level.

About 66% of participants (63.5% women vs. 72.7% men; *p* < 0.001) reported performing weekly PA. Men spent more time performing PA (210.0 vs. 174.3 min/week; *p* = 0.009) and had a higher metabolic rate (2835.0 vs. 1408.2 METs/week; *p* < 0.001). Men preferred running, playing football, and fitness center activities, and women preferred cycling, running, and swimming (data not shown). We asked participants who did not perform PA about the reasons for this. The most common responses were “lack of time” (33.6%), “lack of will” (19.8%), and “schedules of work and lessons” (13.5%) (data not shown). Since starting at the university, about 15% of participants stopped performing PA, 33.0% decreased their PA, and 18.8% increased their PA. Among men, there were negative correlations between the duration of PA and time using a computer (r = –0.324; *p* < 0.01) and hours of studying (r = –0.182; *p* < 0.05). In contrast, among women, there was only a negative correlation between duration of PA and time spent watching television (r = –0.182; *p* < 0.05).

A total of 19.5% of the participants were smokers (17.2% habitual smokers and 2.3% occasional smokers). Social class and sex were unrelated to smoking. Among smokers, individuals consumed an average of 7.1 (SD 7.0) cigarettes per day (Table 2).

A total of 58.0% of the participants were regular consumers of alcohol. Among alcohol consumers, the mean ethanol consumption per week was 47.8 g (SD 61.7), with 15.5 g on weekdays (SD 39.5) and 32.3 g on weekends (SD 32.7). Men consumed more alcohol than women (56.2 (SD 73.8) vs. 34.1 (SD 50.0) g/week; *p* < 0.001). Regarding the consumption of other drugs, the highest weekly consumption was for cannabis 26.5% (3.1% daily), followed by tranquillizers (13.8%), cocaine (5.9%), hallucinogens (4.5%), ecstasy (2.9%), and amphetamines (2.3%).

The mean PREDIMED score was 4.6 (SD 1.5) and the mean MDS was 5.3 (SD 1.8). These two scores had a high and statistically significant correlation (r = 0.709; *p* < 0.01). Women had higher adherence to the MD than men (PREDIMED score: 4.8 vs. 4.1, *p* < 0.001; MDS: 5.5 vs. 4.8, *p* < 0.001). On average, men ate 2.5 meals per day, and women ate 2.7 meals per day (*p* = 0.024). About 63% of participants reported having breakfast every day, and 10.2% of them reported never or almost never having breakfast.

### 3.3. Relationships of Mediterranean Diet and Physical Activity

Table 3 shows the factors associated with PA. Univariate analysis showed that males, those who had greater adherence to the MD, those whose parents had high PA, and those who spent less time studying and using a computer had greater PA. Multivariate analysis which was controlled for all these factors indicated that greater adherence to the MD, spending less time using a computer, and being male remained significantly associated with PA.

Table 4 shows the factors associated with adherence to the MD. Univariate analysis showed that females, those who were older, those who performed more PA, and those who had a high frequency of eating breakfast and ate more meals per day had greater adherence to the MD. Multivariate analysis indicated that these same factors remained associated with MD, and that tobacco consumption was related to a lower adherence to the MD.

### 3.4. Cluster Analysis of Health-Related Lifestyles

Our cluster analysis of the health-related lifestyles indicated the presence of three different clusters (Figure 1). This analysis explained the grouping of 435 participants; nine participants did not belong to any cluster. We characterized these clusters as follows:

Cluster 1 (Unhealthy lifestyles/High risk): Participants had moderate adherence to the MD; high consumption of tobacco but low consumption of alcohol; a low level of PA; and poor quality of life, bad health status, and perception of a high level of stress.

Cluster 2 (Unhealthy lifestyles/Moderate risk): Participants had a low adherence to the MD; high alcohol consumption; moderate tobacco consumption; moderate level of PA; and moderate health status, quality of life, and stress.

Cluster 3 (Healthy lifestyles/Low risk): Participants had a high adherence to the MD; high level of PA; low consumption of tobacco and a moderate-low consumption of alcohol; and a low level of perceived stress, very good quality of life, and good health status.

Our analysis of these clusters indicated there were significant differences according to sex (χ2(2) = 11.149; *p* = 0.040), age (F = 10.284; *p* < 0.001), BMI (χ^2^(2) = 7.559; *p* = 0.023), and living arrangements during the time of university attendance (χ^2^(2) = 19.911; *p* < 0.001), but no significant effect of social class (Table 5). Participants in cluster 1 were mainly female (75.9%) and had a mean age of 23.4 years (SD 5.7). Participants in cluster 2 had an average age of 26.6 years (SD 7.8), had the highest prevalence of obesity and overweight, and mostly lived away from their parents. Participants in cluster 2 and cluster 3 had a mean age of 21.9 years (SD 4.4), mostly lived with their parents, and had the lowest prevalence of obesity and overweight.

## 4. Discussion

Our analysis of the health-related lifestyles of university students, such as PA, diet, tobacco consumption, and alcohol consumption, indicated significant clustering and associations with stress, perceived health status, and quality of life. The present study provides strong evidence that the health-related lifestyles of university students are a serious public health issue, that could contribute to the rise of noncommunicable diseases in the population. 

About two-thirds of our university students reported performing PA, more than in similar studies [8,19,33,34]. We also found that men had greater PA levels than women. Cultural factors and traditions could explain these gender differences. Previous studies suggested that the same-sex parent affects PA of an individual, and other studies reported that the mother’s education level is particularly relevant [35,36]. However, we found that the PA of both parents had no significant effect on the PA of an individual. Furthermore, environmental factors can also influence PA. For example, the movement from the controlled environment of a secondary school to the more independent environment of the university could impact PA patterns [37]. Further research is needed to examine this topic.

Our results regarding the consumption of recreational substances indicated a lower prevalence of smoking (19.5%) than previously reported for the same environment [33,35]. However, our results on the prevalence of alcohol consumption were similar to previous studies, in that about six out of ten university students were regular consumers, most consumption occurred on weekends, and men consumed more than women [5,33,38]. There has been a change in the pattern of alcohol consumption by young people, with decreasing weekday consumption and increasing weekend consumption, as alcohol consumption has become an essential component of the leisure time of young people [39]. The social tolerance and the perception of the low risk associated with drinking alcoholic beverages have contributed to the widespread consumption and normalization of this behavior. Gender differences regarding the social acceptance of alcohol consumption could explain the lower consumption by women [40]. It is not clear how universities should address problematic alcohol consumption by students, and there is little evidence of the effectiveness of commonly used intervention techniques (motivational interviews or social norms campaigns) [41].

The present study showed a clear “westernization” of students’ diets, with a movement away from the traditional MD [42], which consists of olive oil, fruits, nuts, whole-grain breads, legumes, fresh foods, eggs, and fish. University students are likely to adopt fashionable diets and to eat fast food, among other tendencies [43]. As a result, there is an increase in the consumption of processed meats, snacks, butter and margarine, sweets, and pastries (foods with abundant saturated fats and sugars) [35].

Most studies that assessed adherence to the MD showed that university students had low-to-medium adherence [43,44], with lower intake of fruits and vegetables than recommended [8,38]. In agreement with other studies, our results showed a positive correlation of the daily number of meals with adherence to the MD [35]. We also identified healthier eating patterns (higher adherence to the MD) among women, as previously reported [35,45]. For these reasons, we suggest that interventions designed to promote healthy lifestyles in university students should consider gender, because men engage in more PA, but women have greater adherence to the MD.

Krieger et al. [46] reported an inverse relationship between a healthy diet (adherence to the MD) and tobacco consumption, in agreement with the present study. However, other research reported no relationship between diet quality and smoking [47]. In contrast to previous studies that examined PA [8,35,44], our results showed no association between PA and consumption of tobacco, alcohol, and other drugs. The lack of a relationship between alcohol consumption and PA could be because alcohol consumption is deeply rooted in Spanish society, and the MD even encourages the moderate consumption of wine. However, our results showed a positive association between PA and diet quality. Only a few previous studies have evaluated this relationship [35,38,48], and they reported that more active people usually consume more fruits and vegetables than less active people. However, it is unknown whether physiological needs due to increased PA, or cultural, social, or psychological factors influence food selection. Considering the health benefits of the MD and PA and their close relationship, we suggest promotion of a Mediterranean lifestyle (MD, PA, social activities, and moderate sun exposure) for university students.

Our results support the findings of previous studies which showed that health-related lifestyles should not be considered as isolated factors, because healthy lifestyles tend to cluster in the same way as unhealthy lifestyles [18,49]. Three distinct clusters were identified: Cluster 1 that we designated “unhealthy lifestyles/high risk”, and cluster 2 “unhealthy lifestyles/moderate risk”, because we consider these students at a high risk of different chronic diseases. In contrast, cluster 3 “healthy lifestyles/low risk” was comprised by students with low risk of health problems (adequate diet, PA practice, nonsmoking behavior, and less stress). Our cluster analysis demonstrated that more than half of university students had unhealthy behavior patterns and were in the high- or moderate -risk clusters (cluster 1 or 2). On the other hand, low levels of perceived psychological stress correlated with good health and quality of life, a high-quality diet, practice of PA, low tobacco consumption, and low-to-moderate alcohol consumption. These findings are consistent with previous results [8,34,44], and suggest a clear commitment to health by students in the low-risk cluster (cluster 3). In this regard, previous studies also reported that belonging to a high-risk (unhealthy) or a low-risk (healthy) group is associated with tobacco use, PA, alcohol consumption, and diet quality, and that these factors were associated with each other [8,33,35,44]. Considering these results, we suggest the implementation of interventions that simultaneously promote multiple healthy lifestyles, because this approach would have a greater impact on public health than the promotion of isolated lifestyles [49].

Health care and academic authorities should consider the relationship of psychological stress and unhealthy lifestyles when developing interventions for the university environment. Students experience peaks of anxiety during the academic year [33], and may be more prone to unhealthy lifestyles at these times. However, we are unaware of the implementation of measures that counteract the negative effect of these peaks, such as stress management during the exam period. It is noteworthy that a greater percentage of women than men reported feeling stressed. This could partly be due to the longer time women spend engaged in academic activities (attendance at lessons and time studying) [50].

Therefore, when planning policies and designing and evaluating interventions, the behavior patterns within these clusters should be considered [20,49,51]. However, previous studies have not consistently identified the occurrence of these clusters. Although previous studies of general populations identified relationships of socio-economic status with lifestyle [52], we found no such relationship. This may be because all of our participants were university students with a higher educational level. We thus suggest the performance of additional cluster studies of young people outside the university environment that consider socio-economic factors when planning interventions.

### Study Limitations and Strengths

Several limitations of this study should be acknowledged. Data collection was performed using a self-reported survey. Self-reported measures of lifestyles may be subjected to social desirability bias. For example, individuals tend to underestimate body weight and overestimate height [53]. However, previous studies that used self-reported questionnaires support the general validity of this methodology [54]. Furthermore, the current study was limited to cross-sectional data. Therefore, we could not establish casual relationships between factors. However, the applied design was effective in identification of clusters of lifestyles. 

Furthermore, as expected in studies of university students, there were more female than male participants (67.8% vs. 32.2%). This was also reported in previous studies [13,44,55,56]. However, when considering the overall percentages of male and female students, the difference in the present study was greater than expected. It should be considered that all participants in the present study were students attending lessons on a specific day and hour. Since absenteeism is higher among men, this could have affected the sex ratio of participants. There could also be a relationship of absenteeism with certain health-related lifestyles, so we may have underestimated the prevalence of these health-related lifestyles. Although, the participants represented only one university, the context of the University of Balearic Islands is similar to other Spanish universities.

Despite these limitations, the present study provides important information about the clustering of different health-related lifestyles in young adults, and the relationship of these lifestyles with self-perceived stress, health status, and quality of life. Furthermore, we examined variables that have major impacts on health and contribute to mortality and morbidity [3,4]. The clustering of health-related lifestyles among students has not yet been sufficiently recognized, and requires the attention of university administrators. Universities in Spain have formed a partnership in a Network of Healthy Universities [57], and most of these universities have health promotion services and facilities. However, there is still no strong culture of health promotion, and promotion of health is performed without the coordination or collaboration of the health care administration. Universities could provide an environment that better promotes healthy lifestyles [33,58]. For example, universities could provide restaurants that offer a MD with more fruit and vegetable options, promote the use of the fitness centers, design pedestrian-friendly walkways, promote smoke-free and alcohol-free campuses, and provide informative promotional materials to prevent risky lifestyles [59]. In fact, because university students are future parents, teachers, health care professionals, officials, and politicians, these changes can foster a vision towards the promotion of health from all these fields [60]. 

## 5. Conclusions

The results of the present study suggest that there is a need to improve the health-related lifestyles of university students. Furthermore, the health-related lifestyles of students that are related to diet, PA, smoking, and alcohol consumption tended to cluster together. In addition, these factors were related to self-reported stress, health status, and quality of life. These results suggest that health promotion interventions in the university environment should consider gender differences and should simultaneously focus on multiple health-related lifestyles rather than any single behaviour. Similar studies are needed to examine to what extent these findings can be replicated in other universities and which programs are more effective in addressing multiple health-related lifestyles of the students.

## Figures and Tables

**Figure 1 ijerph-17-01776-f001:**
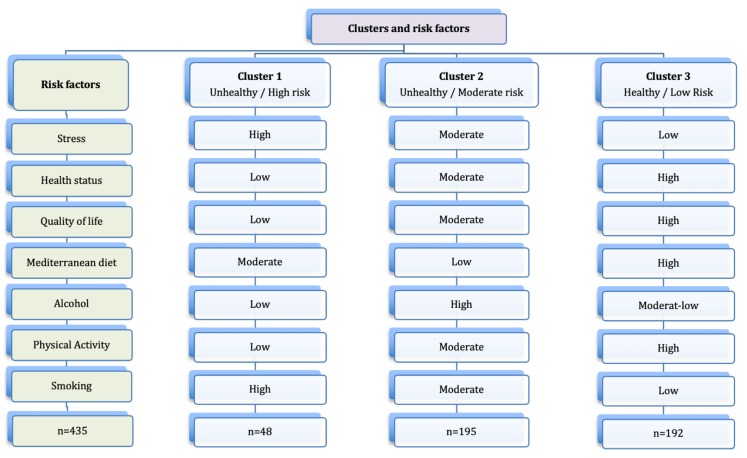
High, moderate, and low risk clusters and their associated risk factors.

**Table 1 ijerph-17-01776-t001:** General characteristics of participants (*n* = 444).

	All (*n* = 444)	Men (*n* = 143)	Women (*n* = 301)	*p-*Value
	Mean (SD) or *n* (%)			
**Age, years**	23.1 (5.7)	23.0 (5.5)	23.0 (5.7)	0.681
**Weight, kg**	64.2 (12.9)	75.8 (11.5)	58.7 (9.4)	<0.001
**Height, cm**	168.5 (8.7)	177.5 (6.6)	154.1 (5.8)	<0.001
**BMI, kg/m^2^**	22.5 (3.5)	24.0 (3.5)	21.8 (3.2)	<0.001
**BMI classification ***				<0.001
Underweight	34 (7.7%)	1 (0.7%)	33 (11.0%)	
Normal weight	326 (73.4%)	101 (70.6%)	225 (74.7%)	
Overweight	68 (15.3%)	35 (24.5%)	33 (11.0%)	
Obese	16 (3.6%)	6 (4.2%)	10 (3.3%)	
**Social class**				0.044
High	105 (23.7%)	44 (30.8%)	61 (20.3%)	
Medium	271 (61.0%)	79 (55.2%)	192 (63.8%)	
Low	68 (15.3%)	20 (14.0%)	48 (15.9%)	
**Working**				
Yes	130 (29.4%)	101 (29.4%)	211 (29.4%)	0.990

* Underweight (BMI < 18.5 kg/m^2^); normal weight (BMI 18.5 to 25 kg/m^2^); overweight (BMI 25 to 30 kg/m^2^); obese (BMI ≥ 30 kg/m^2^).

**Table 2 ijerph-17-01776-t002:** Health-related behaviors of participants (*n* = 444).

	All (*n* = 444)	Men (*n* = 143)	Women (*n* = 301)	*p*-Value
	Mean (SD) or *n* (%)			
**Stress** High or very high	225 (50.6%)	56 (39.2%)	169 (56.7%)	<0.001
**Health status** Good or very good	401 (91.0%)	130 (91.6%)	271 (90.7%)	0.061
**Quality of life** Good or very good	394 (88.9%)	129 (90.2%)	265 (88.4%)	0.711
**Diet**				
Meals/day	2.7 (0.9)	2.5 (1.0)	2.7 (0.9)	0.024
Breakfast every day	279 (63.1%)	89 (62.2%)	190 (63.5%)	0.790
Diet quality				
PREDIMED score	5.4 (1.8)	4.8 (1.8)	5.6 (1.8)	<0.001
MDS	5.3 (1.8)	4.8 (1.9)	5.5 (1.7)	<0.001
**Physical activity**				
Yes	294 (66.2%)	103 (72.0%)	191 (63.5%)	<0.001
Mets-min/week	1867.7 (2798.5)	2835.0 (3991.4)	1408.2 (1834.6)	<0.001
Minutes/week	186.6 (89.1)	210.0 (99.4)	174.3 (80.9)	0.009
**Parent’s physical activity**				0.311
Yes, both	79 (19.7%)	28 (20.9%)	51 (19.0%)	
Yes, mother only	49 (12.2%)	15 (11.2%)	34 (12.7%)	
Yes, father only	63 (15.7%)	15 (11.2%)	48 (17.9%)	
**Smoking**				
Yes	86 (19.5%)	22 (15.5%)	64 (21.3%)	0.148
Cigarettes/day	7.1 (7.0)	8.4 (9.2)	6.6 (6.1)	
**Alcohol consumption**				
Yes	41.2 (59.6%)	56.2 (73.8%)	34.1 (50.0%)	<0.001
g ethanol /week	41.2 (59.6)	56.2 (73.8)	34.1 (50.0)	<0.001
**Drug consumption**				
Yes	65 (14.6%)	18 (12.6%)	47 (15.6%)	0.399
**Sedentary habits**				
Study (h/day)	3.5 (2.3)	3.3 (3.4)	3.6 (2.4)	0.047
Computer (h/day)	3.6 (2.5)	3.8 (2.6)	3.6 (2.4)	0.694

**Table 3 ijerph-17-01776-t003:** Univariate and multivariate analysis of factors associated with physical activity ^†^ (*n* = 444).

Physical Activity	Yes (*n* = 294)	No (*n* = 150)	OR	95% CI	aOR	95% CI
	*n* (%) or Mean (SD)					
**Gender**						
Male	105 (35.7%)	38 (25.7%)	1		1	
Female	191 (65.0%)	110 (73.3%)	0.62 *	0.40–0.96	0.49 *	0.29–0.85
**Age (years)**	22.9 (5.6)	23.3 (5.7)	0.99	0.96–1.02	0.99	0.95–1.04
**BMI (kg/m^2^)**	22.6 (3.1)	22.4 (4.2)	1.01	0.96–1.07	1.02	0.95–1.09
**Social class**						
High	72 (24.5%)	35 (23.3%)	1			
Medium	83 (28.2%)	43 (28.7%)	1.07	0.65–1.77		
Low	139 (47.3%)	72 (48.0%)	1.01	0.63–1.62		
**Mediterranean Diet Score**	5.5 (1.8)	5.0 (1.7)	1.13 *	1.01–1.26	1.16 *	1.02–1.32
**Parent’s physical activity**						
No	143 (48.6%)	89 (59.3%)	1		1	
Yes, both	69 (23.5%)	19 (12.7%)	2.24 **	1.24–4.06	1.82	0.96–3.47
Yes, mother only	39 (13.3%)	16 (10.7%)	1.50	0.77–2.92	1.78	0.83–3.82
Yes, father only	43 (14.6%)	26 (17.3%)	1.00	0.57–1.79	0.84	0.45–1.55
**Computer use (h/day)**	3.1 (2.1)	3.8 (2.2)	0.87 ***	0.80–0.96	0.85 **	0.76–0.95
**Studying (h/day)**	3.3 (2.0)	3.7 (2.0)	0.89 *	0.81–0.98	0.92	0.81–1.04
**Tobacco consumption**						
No	236 (80.3%)	(80.8%)	1			
Yes	58 (19.7%)	(19.2%)	1.02	0.62–1.69		
**Alcohol consumption**						
No	39 (13.3%)	29 (19.3%)	1			
Yes	255 (86.7%)	121 (80.7%)	1.54	0.91–2.61		
**Drug consumption**						
No	243 (82.7%)	121 (80.7%)	1			
Yes	51 (17.3%)	29 (19.3%)	0.90	0.54–1.49		

* *p* <0.05, ** *p* <0.01, *** *p* <0.001. ^†^ Physical activity (yes vs. no) was calculated excluding the “walking” category. Area under the curve (AUC) of the logistic regression model was 0.68. Qualitative explanatory variables that had more than two categories were transformed into dummy variables for inclusion in the logistic model.

**Table 4 ijerph-17-01776-t004:** Univariate and multivariate analysis of factors associated with adherence to the Mediterranean diet (*n* = 431).

Mediterranean Diet Score (MDS)	Low Adherence(MDS < 5)(*n* = 229)	High Adherence(MDS ≥ 5)(*n* = 202)	OR	95% CI	aOR	95% CI
	*n* (%) or Mean (SD)					
**Sex**						
Male	104 (45.4%)	54 (26.7%)	1		1	
Female	125 (54.6%)	148 (73.3%)	2.26 ***	1.47–3.47	2.98 ***	1.80–4.93
**Age (years)**	21.9 (4.4)	23.4 (5.9)	1.06 *	1.01–1.11	1.08 **	1.02–1.14
**BMI (kg/m^2^)**	22.4 (3.8)	22.6 (3.4)	1.02	0.96–1.08	1.05	0.98–1.13
**Social class**						
High social class	51 (22.3%)	51 (25.2%)	1			
Medium social class	73 (31.9%)	53 (26.2%)	0.71	0.40–1.27		
Low social class	105 (45.8%)	98 (48.6%)	0.93	0.54–1.58		
**Physical activity ^†^**						
No	95 (41.5%)	62 (30.7%)	1		1	
Yes	134 (58.5%)	140 (69.3%)	1.61 *	1.05–2.47	1.75 *	1.09–2.80
**Breakfast every day**						
No	114 (49.8%)	62 (30.7%)	1		1	
Yes	115 (50.2%)	140 (69.3%)	2.24 ***	1.46–3.42	1.70 *	1.05–2.76
**Meals/day**	2.4 (0.9)	2.8 (0.9)	1.61 ***	1.27–2.05	1.43 **	1.10–1.87
**Tobacco consumption**						
No	173 (75.5%)	168 (83.2%)	1		1	
Yes	56 (24.5%)	34 (16.8%)	0.61	0.37–1.01	0.52*	0.30–0.91
**Alcohol consumption**						
No	32 (14.0%)	31 (15.4%)	1			
Yes	197 (86.0%)	171 (84.6%)	0.89	0.49–1.60		
**Drugs consumption**						
No	185 (80.8%)	168 (83.2%)	1			
Yes	44 (19.2%)	34 (16.8%)	0.86	0.50–1.46		

* *p* < 0.05, ** *p* < 0.01, *** *p* < 0.001. ^†^ Physical activity (yes vs. no) was calculated excluding the “walking” category. AUC of the logistic regression model was 0.67. Qualitative explanatory variables that had more than two categories were transformed into dummy variables for inclusion in the logistic model.

**Table 5 ijerph-17-01776-t005:** Characteristics of the three clusters (*n* = 435).

	Cluster 1 (*n* = 195, 44.83%) Unhealthy, High Risk	Cluster 2 (*n* = 48, 11.03%) Unhealthy, Moderate Risk	Cluster 3 (*n* = 192, 44.14%) Healthy, Low Risk
**Health-Related Behaviors**	***n*** **(%) or Mean (SD)**		
**Stress ****	4.14 (0.76)	3.58 (1.07)	2.78 (0.89)
**Health status ****	1.18 (0.53)	0.83 (0.43)	0.47 (0.52)
**Quality of life ****	1.20 (0.58)	0.98 (0.53)	0.60 (0.51)
**Mediterranean diet**	5.34 (1.74)	4.82 (1.56)	5.42 (1.83)
**Physical activity ****	1071 (1519)	2086 (2549)	2473 (2547)
**Smoking ****	37 (19.0%)	32 (66.7%)	15 (7.9%)
**Alcohol consumption****	30.05 (2.15)	109.66 (15.83)	35.23 (2.54)
**Sociodemographic factors**			
**Age ****	23.4 (5.7)	26.6 (7.8)	21.9 (4.4)
**Gender ***			
Female	148 (75.9%)	29 (60.4%)	117 (60.9%)
Male	47 (24.1%)	19 (39.6%)	75 (39.1%)
**BMI ***			
Normal weight	150 (77.7%)	36 (75.0%)	166 (87.4%)
Obesity or overweight	43 (22.3%)	12 (25.0%)	24 (12.6%)
**Social class**			
Blue collar	96 (50.8%)	26 (56.5%)	80 (43.0%)
White collar	93 (49.2%)	20 (43.5%)	106 (57.0%)
**Residence ****			
Parents	130 (66.7%)	20 (42.6%)	146 (76.0%)
Others	65 (33.3%)	27 (57.4%)	46 (24.0%)

* *p* < 0.05, ** *p* < 0.01, *** *p* < 0.001.

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
