# Peer review of "Cluster Analysis of Health-Related Lifestyles in University Students"

_ijerph, 2020, doi:10.3390/ijerph17051776_

Round 1
Reviewer 1 Report
Overall comments: Overall this is a nicely done study and the report is well-written. It was a pleasure to read. There were several things that I particularly liked about the study, especially the results section. The use of no more that 2 decimal places when reporting numbers makes the tables so much easier to read. Also, I liked your use of a sentence at the beginning of the paragraph interpreting the results stating which table you were referring to (sometimes referred to as meta-discourse and is very helpful to the reader). Hence, my comments are related to a few issues that you can consider.
It would be helpful to investigators who may want to adapt your methods to provide citations to the Spanish National Health Survey and “other validated measures” (lines 90 & 91). Line 135—sweets are listed twice. A bit more description of the 8 typical and 2 atypical MD food groups would be helpful to the reader. Also, line 142, can you provide a citation for the listwise deletion method?
Related to the reporting of the clusters, it seems like a picky comment but I think it would be very helpful to the reader to order the clusters from high risk to low risk or vice versa. That way it is easier to see the progression along a continuum. Also consider more discussion of the clusters. The one finding that really jumped out to me is the size of the unhealthy/high risk group which was about 45% of the sample and that they report low health status. Does this mean that almost half of university students have some type of chronic disease? Perhaps you could discuss more as I could not think of other reasons for poor health status. Several years ago, I surveyed professional pharmacy students at our institution and found the between 10 and 15% had chronic pain. There are lots of other chronic diseases so if one included them all then perhaps it would be around 40%. If this is so, then that has major implications for student services in university settings. The low health status of this group could also be related to their lack of exercise, stress, and other lifestyle factors. So could you discuss more?
Again, a very nicely done study and the report is well-written.
Author Response
Dear Iulia-Elena Neacsu,
Please find enclosed the revised version of our manuscript entitled “Cluster analysis of health-related lifestyles in university students” (IJERPH-725583) and a point-by-point reply to all the comments raised by the reviewers, to whom we are grateful. We have carefully revised our manuscript taking into account comments and suggestions of reviewers. And, the whole manuscript has been reviewed and corrected for English grammar errors.
Thank you very much for your time and interest.
We look forward to hearing from you soon.
Sincerely,
Miquel Bennasar-Veny
Response to Reviewer 1 Comments
Point 1: Overall this is a nicely done study and the report is well-written. It was a pleasure to read. There were several things that I particularly liked about the study, especially the results section. The use of no more that 2 decimal places when reporting numbers makes the tables so much easier to read. Also, I liked your use of a sentence at the beginning of the paragraph interpreting the results stating which table you were referring to (sometimes referred to as meta-discourse and is very helpful to the reader). Hence, my comments are related to a few issues that you can consider.
It would be helpful to investigators who may want to adapt your methods to provide citations to the Spanish National Health Survey and “other validated measures” (lines 90 & 91).
Response 1: Thank you ever so much for your comments. Following your suggestions, we have provided a new reference for the Spanish National Health Survey. In order to clarify “other validated measures”, we have added all the references at the end of the sentence.
Point 2: Line 135—sweets are listed twice.
Response 2: Sorry for the mistake, we have amended it.
Point 3: A bit more description of the 8 typical and 2 atypical MD food groups would be helpful to the reader.
Response 3: We agree with the reviewer and a description of the typical and atypical MD food groups has been added (page 3 lines 139-140).
Point 4: Also, line 142, can you provide a citation for the listwise deletion method?
Response 4: Thanks for the suggestion, we have provided a new reference (Eekhout et al., 2012) for the listwise deletion method.
Point 5: Related to the reporting of the clusters, it seems like a picky comment but I think it would be very helpful to the reader to order the clusters from high risk to low risk or vice versa. That way it is easier to see the progression along a continuum.
Response 5: Many thanks for this clarification, we agree with the reviewer. We have changed the order of the clusters from high risk to low risk in the text, in the figure 1 and in the table 5.
Point 6: Also consider more discussion of the clusters.
Response 6: Following your suggestion we have added more discussion of the clusters in the discussion section (page 10, lines 328-332).
Point 7: The one finding that really jumped out to me is the size of the unhealthy/high risk group which was about 45% of the sample and that they report low health status. Does this mean that almost half of university students have some type of chronic disease? Perhaps you could discuss more as I could not think of other reasons for poor health status. Several years ago, I surveyed professional pharmacy students at our institution and found the between 10 and 15% had chronic pain. There are lots of other chronic diseases so if one included them all then perhaps it would be around 40%. If this is so, then that has major implications for student services in university settings. The low health status of this group could also be related to their lack of exercise, stress, and other lifestyle factors. So could you discuss more?
Response 7: Thank you for your comment. The health status variable in our study was categorized only for descriptive purposes (table 2). But, in the cluster analysis the variables were used as originally collected in 5 categories (“very good”, “good”, “fair”, “bad”, or “very bad”). Although, cluster 1 is a group with the worst perception of health, the mean health status punctuation is 1.18 that is between “good” and “fair” responses. In this sense we cannot consider that the 45% of the sample reported a low health status.
We have clarified the categorization of this variable in the methods section (page 3, line 104).
Reviewer 2 Report
Well written and referenced manuscript with a reasonable number of participants. Study limitations could also include that the participants represented only one university. The manuscript could be enhanced by an introduction that would describe the context of the University of Balearic Islands. The demographics of the university, nature of the students of who attend the university along with the university profile would help the reader understand the context. In reading the findings, it would then help in understanding those that made the decision to participate. Otherwise the section on strength and limitations of the study was adequate.
The conclusion section is a little strong given that the participants represent only one university. The study findings are not generalizable to all college age students. A better recommendation might be that this study be replicated in other universities. In addition the conclusion section could be developed further.
Author Response
Dear Iulia-Elena Neacsu,
Please find enclosed the revised version of our manuscript entitled “Cluster analysis of health-related lifestyles in university students” (IJERPH-725583) and a point-by-point reply to all the comments raised by the reviewers, to whom we are grateful. We have carefully revised our manuscript taking into account comments and suggestions of reviewers. And, the whole manuscript has been reviewed and corrected for English grammar errors.
Thank you very much for your time and interest.
We look forward to hearing from you soon.
Sincerely,
Miquel Bennasar-Veny
Response to Reviewer 2 Comments
Point 1: Well written and referenced manuscript with a reasonable number of participants. Study limitations could also include that the participants represented only one university. The manuscript could be enhanced by an introduction that would describe the context of the University of Balearic Islands. The demographics of the university, nature of the students of who attend the university along with the university profile would help the reader understand the context. In reading the findings, it would then help in understanding those that made the decision to participate. Otherwise the section on strength and limitations of the study was adequate.
Response 1: Thank you ever so much for your comments. We have now included in the limitations section that the participants represented only one university (University of Balearic Islands). The context of the University of the Balearic Islands is similar to other Spanish universities (page 11, lines 372-373).
Point 2: The conclusion section is a little strong given that the participants represent only one university. The study findings are not generalizable to all college age students. A better recommendation might be that this study be replicated in other universities. In addition the conclusion section could be developed further.
Response 2: We appreciate this recommendation, thank you very much. The conclusion section has been revised, and now included your suggestions (page 12, lines 396 – 399).